# Lesbian, gay, bisexual, transgender, intersex, and its legalisation in Africa: Insights from tertiary-level students in Ghana

Francis Acquah [ID]*[◎], Charles Owusu-Aduomi Botchwey[◎], Prince Owusu Adoma[◎], Emmanuel Kumah[◎]

Department of Health Administration and Education, Faculty of Science Education, University of Education, Winneba, Central Region, Ghana

◎ These authors contributed equally to this work.
* nanaacquah1122@gmail.com

**Data Availability Statement:** All relevant data are within the paper and its Supporting information files.

## Abstract

### Introduction

Lesbian, gay, bisexual, transgender, and intersex (LGBTI) and related activities have been a topic of debate and discussion among policymakers and stakeholders, as well as common citizens in the African region, especially in Ghana. The current anti-LGBTI-related bill being put before Ghana's Parliament signifies the intensity of the issue. Even though some studies have looked at some aspects of the issue, no study presently has explored people's opinion on the passage of any future anti-LGBTI and related legislations in Ghana.

### Aim

This study examined the perspective of tertiary-level students on the passage of anti-LGBTI legislation, as well as the non-physical factors that influence support for the passage of anti-LGBTI and related legislation in Ghana.

### Methods

The study employed a quantitative cross-sectional design using 1,001 tertiary-level students. The study used convenience sampling technique with an online closed-ended, structured survey questionnaire as the main data collection instrument. The data was then analysed using Statistical Package for the Social Sciences, version 29 at a 5% significance level.

### Results

The results of the study indicated that majority of the respondents (81%) were in support of the passage of anti-LGBTI and related legislations. Their reasons included the health implications of LGBTI and related activities (63%), cultural and societal values (62%), religious reasons (54%), and western culture (25%). Also, almost half of the respondents (49%) held that health related perceptions about LGBTI have little or no empirical basis. The inferential analysis, further, revealed that even when age and sex assigned at birth are controlled,

**Funding:** The authors received no funding for this work.

**Competing interests:** The authors have stated that there are no competing interests.

perceived health implications of LGBTI (β = 0.247, p = < .001), religious beliefs (β = 0.189, p = < .001), and cultural values (β = 0.218, p = < .001) positively predict the support for passage of anti-LGBTI legislation.

## Conclusions

People's level of support for the passage of anti-LGBTI legislation is influenced by several factors including religious beliefs, cultural values, and the perceived health implications of LGBTI. There is, however, the need for policy makers and other stakeholders to create awareness and educate the public about the various perceptions about LGBTI and related activities that are not scientifically legitimate.

## 1. Introduction

Democracy as a system of government is said to be characterised by elements such as equality, bills of rights, human rights, the rule of law, and tolerance, among others [1]. However, it appears that when it comes to its practice, some of these elements are not strictly adhered to both in developed and in developing democratic countries, taking into consideration the issue of equality and discrimination, for instance.

Whilst there has been the clarion call to significantly ensure equality and non-discrimination among individuals despite their race, gender, ethnicity, religion, and sexual orientation by various authorities and human rights activists, people are still discriminated against based on their characteristics [2]. Among the groups of people who are faced with a significant level of discrimination in almost all societies around the world are lesbian, gay, bisexual, transgender, and intersex (LGBTI) people [3]. Also often generally referred to as Lesbian, Gay, Bisexual, Transgender/Transsexual plus (LGBT+). These people are often unfairly treated because of how people perceive them for expressing their sexual feelings towards their fellow gender. In some countries, such act is perceived to be uncommon and alien to their culture, thereby, going to the extent of criminalising such persons' statuses and related activities. This is often aided by the passage of anti-LGBTI and related laws in the various countries. Anti-LGBTI and related laws are laws specifically imposed on LGBTI and their related activities that often tend to promote discrimination and violence against LGBTI. According to Statista [4], more than 68 countries have criminalised homosexuality, with 11 countries even going to the extent of imposing death penalties on their related activities.

Even though it is very cumbersome to justify any form of inequality and discrimination against people in this current aeon, the unfair treatment meted out to this minority group of people across the globe often appears to be justified in even democratic states with reference to legislation specially meted out against them. The criminalisation and discrimination against LGBTI people are not a current development and can be traced back even to the early 1860s [5] with its justification spanning beyond religion and morality [6, 7]. These basics for its justification together with other cultural and societal values are still mentioned today by people and authorities who think LGBTI and related activities must not be entertained in our World, taking the speeches on the matter by some African leaders, for example. The statement made by the president of Uganda, Museveni, when asked about the passage of a bill against LGBTI sums it all "respect African societies and their values, if you don't agree just keep quiet just manage our society the way we see, if we are wrong we shall find out by ourselves" [8].

Despite these arguments, pressure by human rights activists has managed to counter the grounds for which LGBTI is criminalised, resulting in many developed and developing democratic countries reshaping their legal orientation toward LGBTI laws and beginning to embrace LGBT+ equality. In the year 2020, the European Commission, for instance, presented its first-ever European Union (EU) Strategy for LGBT+ equality [9] with Article 21 of the EU Charter of Fundamental Rights explicitly prohibiting discrimination based on sexual orientation. In the United States (US), whilst some states have exclusive protection for discrimination against LGBT+, some LGBT+ Americans are not fully protected from discrimination in 27 States as there are no explicit protective laws [10].

It is reported that in Asia, there exists still a significant challenge with regard to criminalisation and discrimination against LGBT+ people, especially in the Middle East [11, 12]. Even though some of the thirty-three African countries that used to have laws against LGBTI, such as Botswana, Angola, Cape Verde, Gabon, and other Portuguese former colonies, have decided to decriminalise LGBTI, general reports indicate that LGBTI individuals still suffer a significant level of discrimination in the African region [13]. A popular reason for opposing LGBTI is the claim that LGBTI is not African but a "Western thing". Such a claim has however, been challenged by some studies and African scholars on the grounds that African culture is no stranger to LGBTI activities such as homosexual behaviours and acts [14–16]. Alimi [17], in refuting the argument of western notion posits that "a culture that celebrates diversity, promotes equality and acceptance, and recognises the contribution of everyone, whatever their sexuality" cannot be a reason for criminalising and discriminating against some minority in the society and therefore those who hold the argument that LGBTI is not an "African thing" do not actually know their history.

## 1.1 LGBTI and related legal issues in Ghana

Ghana, the country that this study focused on, has, in recent years, been recognised to initiate vocal attacks against LGBTI people [13]. The current national debates about the legality and morality of LGBTI and related activities, as well as the deliberation of a bill that may criminalise homosexuality and make advocating for LGBTI a crime by the country's parliament [18] are all indications of the country's concern about LGBTI and its related activities. Oppong [19] was however, right when he pointed out some misleading and erroneous information given by some scholars concerning the state of affairs of LGBTI and its related activities harsh legislations in the country. As indicated by Oppong, the only legislation pertaining to an aspect of LGBTI (specifically homosexuality) for which offenders are guilty of misdemeanor in Ghana is the Criminal Code of 1960 (Act 29) passed in 1960 [19]. The Article 105 of the Act was amended in 2003. As to whether the Act can be applied to all manner of same-sex sexual intercourse, is still a debate among the authorities.

Aside this, no legislations concerning LGBTI and its related activities have been passed since then, until the current 2021 proposal being filed before Parliament for considerations. The bill, otherwise known as the "promotion of proper human sexual rights and Ghanaian family values bill" is said to aim at heavily penalising homosexuality. It is also acknowledged that when finally enacted as law, it may have the potential to allow for a decade of imprisonment for LGBTI individuals, penalising LGBTI activists and punishing publications that would promote any form of homosexuality [18].

As it stands, LGBTI-related activities such as kissing, affection towards same sex, sexual activities between or among women basically (as it does not meet the "penetration" requirement of the section 99 of the Criminal Offences Act) have no legal implications. Only "consensual same-sex relations between two men is a criminal offence within the

meaning of the Criminal Offences Act of 1960" [20]. Even though same-sex intercourse, as characterised by the law as "unnatural carnal knowledge" for men is clearly illegal, identification of oneself as a homosexual is not a legal offence within the legal frameworks of Ghana [20].

Despite the lack of comprehensive legislation against LGBTI individuals in Ghana, it is also relevant to point out the very manner in which LGBTI individuals in the country are often treated by some citizens and leaders. The current pressure by the religious groups on law makers to make legislations against LGBTI and their related activities cannot be underestimated. Same pressure is true for some traditional authorities, with some traditional leaders going to an extreme end of banishing individuals from their communities because of their gay or lesbian status [21]. The incidence in the Western part of Ghana in which the Regional Minister ordered law enforcement agencies to arrest "all gay men and lesbians in the west of the country, and called on landlords and tenants to report anyone they suspected of being gay or lesbian" [22] as well as the "arrest and alleged arbitrary detention of human right activists defending the rights of lesbian, gay, bisexual, trans and gender diverse (LGBT) community" [23] are all few reported incidences that reflect the way and manner in which the LGBTI individuals in practice are being treated by some citizenry, political, religious and traditional leaders in a democratic state. Reasons often cited for these actions taken against LGBTI and related issues in media platform discussions include preservation of societal and cultural values and adherence to religious beliefs.

Whilst it is one thing to dislike something and another to discriminate against it, any form of violence and discrimination against any group of individuals are generally universally not acceptable [24]. Even so, the justification and legalities concerning these incidences against LGBTI individuals in the country have been an altercation among the various authorities and stakeholders of the law.

That notwithstanding, a polarising view of the current debate in Ghana implies that condemning a bill that discriminates against a minority is the same as condemning a bill that promotes Ghanaian cultural values. The two positions, instead, are not necessarily in opposition to each other. Again, whilst it is crucial as a people to preserve and protect our values and culture, there is the need to make the decision rule very clear. If one intends to address issues of cultural and societal values, then one must do so in a comprehensive and fair manner without any favour and external influence. One cannot in one-instance embrace things that are contradictory and inimical to one's values and culture because it is in favour of some influential people, and criminalise similar things because in this case they have to do with some minority with no or little influence. For instance, if for cultural and procreational reasons, one wants to make policies that deter deliberate attempt of individuals not to give birth, then such legislation must be generalised to everyone. An element of discrimination may be argued to be induced when one wants to punish others and ignore others for the same or similar activity deemed a threat to one's cultural values.

Furthermore, the incessant direct call by the dominant religious leaders in Ghana on the government to always act in a manner to ensure that the core precepts of their religion are adhered to by restricting LGBTI-related activities in the country seems to also imply that all the citizens are religious. This of course is not the case in reality. The argument then is: in a democratic state that respects the rule of law and is open to different shades of belief systems as indicated in Article 1 and 21 of the 1992 Constitution of Ghana [25], to what extent would it be appropriate to make legislation on religious grounds? Considering the core principles of democracy, one can perhaps posit that if there are neutral grounds to pass a legislation in a democratic state, the general idea of protecting and respecting the collective interest and fundamental human rights of all people as enshrined in the 1992 Constitution of Ghana [25] and

the Universal Declaration of Human Rights of the United Nations [26] which the world has decided to show commitment to could be argued to be a laudable, possible guide.

### 1.2 The present study

Given the incessant nationwide debates among academics, media, religious groups and politicians on LGBTI legislation, it has become necessary to explore the views of the individuals looking at the controversial nature of the issue and also account for the factors that actually shape people's opinions on the matter in order to broaden the scope of understanding and discussions on the matter, inform policy makers and other stakeholders given the very limited and predominantly qualitative studies [7, 19, 27, 28] conducted in only some aspects of LGBTI issues in the country (Ghana). Public opinion plays a crucial role in enacting legislation, particularly in a democratic State, and in the discussion of social issues in context. The absence of a study specifically on people's opinions on the passage of LGBTI legislation during a period when LGBTI legislation issues in the country and the African region are at its peak is a gap in the literature that ought to be filled. Moreover, anecdotal evidence suggests that religious, cultural, and health risks are often cited as the rationale for LGBTI legislation by authorities and on various social media platforms [8, 21, 29], yet no empirical studies have been carried out to confirm this anecdotal evidence. A study that addresses this knowledge gap pertaining to people's opinions on LGBTI legislation as well as the driving factors that influence support for LGBTI and related legislation in the country is therefore required.

Since surveying all the people in the country is subjected to limited resources, and on the other hand, the fact that surveying all the population may not be appropriate or feasible due to the nature of the issue in context, studying some portion of the population has been the option. For instance, some local communities consider LGBTI and related activities as an abomination and a taboo and therefore may not want to hear or discuss anything related to it, as can be inferred from Prof. Nii Odaifio III, the Chief and President of "Nungua" Traditional Council in Accra, the capital city of Ghana statement that "There is something we called a taboo which is LGBTQ. . .." [29]. In view of this, tertiary-level students who are more likely to be tolerable and abreast with the current LGBTI issues are considered more appropriate for the study. The few empirical studies conducted in the country looking at people's perceptions and attitudes towards homosexuals, for instance, used tertiary-level students as the study population [30–32]. The insights from the tertiary-level students on the passage of anti-LGBTI and related legislation in the country could thus, be deemed a reliable source of information for policy makers and stakeholders since tertiary-level students are considered among the well-informed in the society.

In summary, the focus of the present study is aimed at exploring the views of the educated, perhaps the most tolerable and probably the immediate future leaders of the country on the passage of any future anti-LGBTI and related activities legislation in the country. Specifically, the study aims at examining the perspectives of tertiary-level students on the passage of anti-LGBTI and related legislation as well as identifying the non-physical factors that can influence the support for the passage of anti-LGBTI legislation in Ghana.

## 2. Theoretical and conceptual background of the study

People's behaviours towards a social phenomenon are influenced by several factors that may extend beyond the dictates of the immediate environment. Even though human behaviours are said to be in a direction that factors in the well-being and interests of others [33, 34], that may only be true to some extent, especially in the African context, where sometimes people are

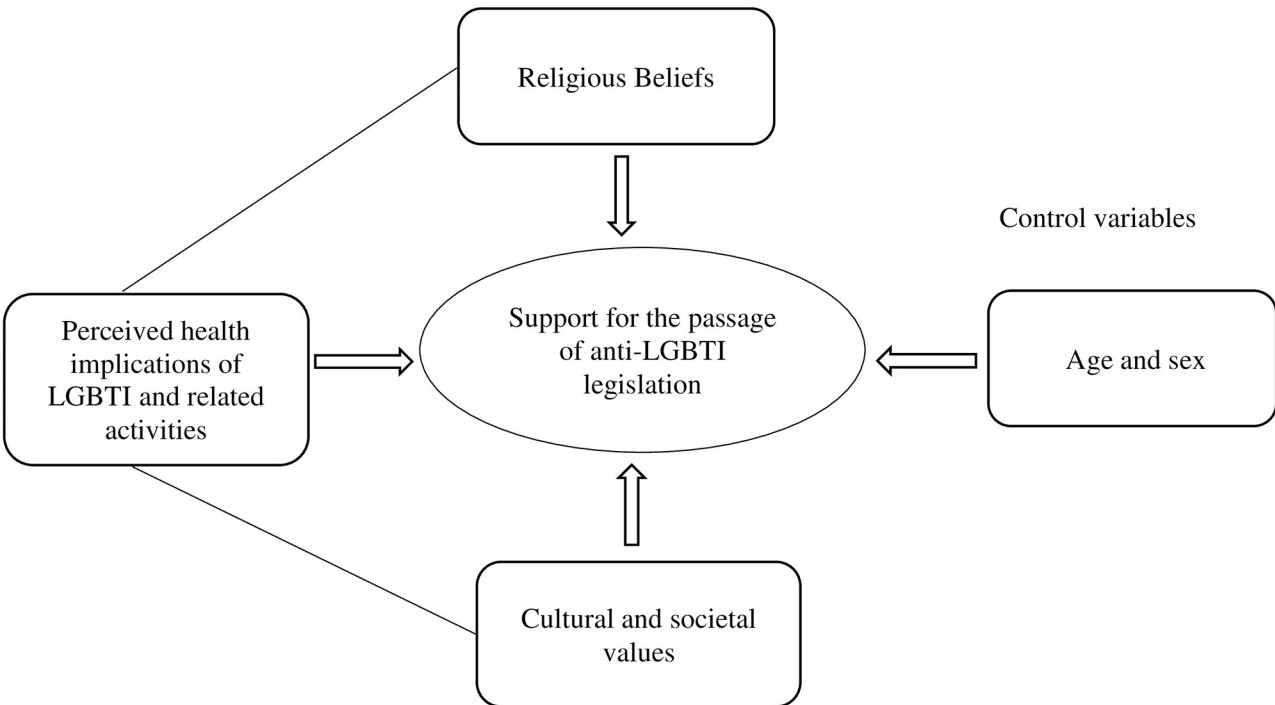

**Fig 1. A conceptual framework depicting the non-physical factors that influence the support for the passage of anti-LGBTI legislation.** Source: Authors own construct, 2023.

challenged or compelled to act in a certain direction devoid of logical and empirical considerations so as to preserve and protect the status quo.

The conceptual framework of this study is therefore underpinned by the African Heritage Theory, which postulates that there exists a real distinction between African culture and other cultures [35] and that there are certain things that are not acceptable and are characterised as not being within the boundaries of African culture which can be traced back to West African culture. The framework of the study also recognises the Health Belief Model, which acknowledges how people are likely to take actions based on their perceived severity about supposed health-related risk activities [36] and finally, the Religious Concept of Sin, which appears to demoralise any activity that is said to be a "sin" in the religious books. Controlling for physical factors such as age and sex, the conceptual framework (Fig 1) of this study thus identifies people's religious beliefs, which are informed by the religious concept of sin as described in the supposed "holy books", societal and cultural values, which according to the African Heritage theory are unique and should be preserved, and perceived health implications, which from the lenses of the health belief model are likely to move people to action, as the main potential predictors of support for the passage of any anti-LGBTI and related legislation at any given point in time.

## 3. Materials and methods

### 3.1 Study design

The study used a quantitative cross-sectional survey design. As in any quantitative study, the focus was to achieve a greater in-depth knowledge about the phenomenon of interest through the collection of numerical data in order to apply multivariate statistics [37]. The present study

adopted a quantitative approach to explore the views of tertiary-level students concerning the passage of any form of anti-LGBTI legislation in Ghana. Since our aim was not to manipulate any independent variable in the study, a cross-sectional study was appropriate to make inferences about the study objectives. This approach was also necessary, looking at the nature and scope of the study population [38].

## 3.2 Population, sample size and sampling techniques

Participants in the study were tertiary-level students in Ghana living in the entire middle and southern parts of the country to ensure that the study captures, if not all, most of the different ethnic groups representation given available resources. Since the exact number of students of the target population was unknown, the "384 magic number" sample size of the KM table proposed by Krejcie and Morgan [39] for a large population was the initial baseline sample size reference for the study. However, taking other factors into consideration, such as the need to reduce the margin of error and the sampling technique employed, the total sample size was significantly increased and estimated at 1,068 using the popular social science online sample size calculator from Calculator.net [40] with a 3% margin of error, a 95% confidence interval (z = 1.96), population proportion of 50% and an unlimited or unknown population size. According to Memon et al. [41], Krejcie and Morgan table and Calculator.net are among the most widely known used sample size estimators among behavioural and social science researchers. The Calculator.net sample size determination calculations for the unknown population of the study were as follows:

$$\text{Sample size for unknown population} =$$

$$n = \frac{Z^2 \times p(1-p)}{\varepsilon^2} \qquad n = \frac{1.96^2 \times 0.5(1-0.5)}{0.03^2} = 1,068$$

Where:
n is the sample size;
z is the z-score for 95 percent confidence interval;
p is the population proportion and
$\varepsilon$ is the margin of error.

The study used a convenience sampling technique with an online closed-ended structured survey questionnaire as the main data collection instrument. Since all students in Ghana are familiar with English (English is the language of instruction from basic to tertiary in the country's education system), the questionnaire was administered in English. A conscious effort was also made to order the questionnaire items to ensure that order effects were significantly reduced. Considering the target population, we preferred to use an online sampling technique because it was easier and more efficient as it encouraged many more students to respond freely to an in-person technique in any given time period. Respondents were contacted through a link sent through their various WhatsApp platform across the central and southern parts of the country. The questionnaire was programmed using Google Forms. Respondents were properly informed beforehand about the study aims and anonymity. The data collection began on the 1st of December, 2022 and ended on the 22nd December, 2022. A total of 1,001 respondents voluntarily accepted to take part in the study, reaching 94% of the estimated sample size.

## 3.3 Variables and study measurement

The development of all the items used in the questionnaire to measure the study variables were done in consultation with experts in questionnaire design and in the field of study. The questionnaire was initially tested among 13 students of the University of Education Winneba, who

were also encouraged to comment on the instrument. Their feedback was finally used to improve the structure and wording of the items.

**3.3.1 Support for the passage of anti-LGBTI legislation.** The dependent variable in the study was the degree of support for the passage of anti-LGBTI legislation. This was assessed through a set of six items (See S1 Appendix). Responses were provided on a 7-point scale, ranging from 1 (strongly disagree) to 7 (strongly agree). A sample item was: "In general, to what extent do you agree or disagree with the passage of laws against LGBTI individuals in the country?" A composite measure of the degree of support was computed by averaging the six items and the result is a reliability score of $\alpha = 0.89$.

**3.3.2 Predictors.** The study included a series of predictors of support namely: religious beliefs; cultural values; and perceived health implications. We also controlled for demographic variables, such as sex and age.

*Religious beliefs*. Religious beliefs were measured using four items 7-point Likert scale response questions ranging from 1 (strongly unlikely) to 7 (strongly likely). All items were reversed coded since lower rating inferred support for the passage of anti-LGBTI legislation. Items commenced with the question "Is your religious denomination likely to warmly accept LGBTI people" and ended with "To what extent is identified LGBTI individual likely to become a leader in your denomination". An overall $\alpha = 0.85$ was obtained as a reliability score for the four items.

*Cultural values*. Cultural values were also assessed with four items. Responses were provided on a 7-point Likert scale, ranging from 1 (strongly unlikely) to 7 (strongly likely). Lower ratings reflect support for the passage of anti-LGBTI legislation hence all the items were reverse-coded. Item samples include "LGBTI people are likely to reveal themselves freely in your community" and "To what extent do you think LGBTI-related activities (e.g., same-sex marriage) are likely to be welcomed in your community?" The composite score of the items yielded a reliability score of $\alpha = 0.79$.

*Perceived health implications of LGBTI activities*. This predictor was assessed with four items 7-point Likert scale response questions rated from 1 (strongly unlikely) to 7 (strongly likely). Unlike the others, none of the four items was reverse-coded, as higher ratings inferred a higher degree of support for the passage of anti-LGBTI legislation. There should be passage of laws against LGBTI because; "Sexual intercourse between homosexuals is riskier than sexual intercourse between the opposite sexes in contracting sexual related diseases" and "Homosexuals are mostly less healthy than heterosexuals. (Healthy used here means having good physical and mental condition)" were among the list of sample items. The four items yielded $\alpha = 0.97$ denoting the highest level of reliability among all the measured latent variables in the study.

## 3.4 Data analysis

Data generated from the online survey questionnaire were first imported from Google Forms to Microsoft excel 2013 and then to Statistical Package for the Social Sciences (SPSS) version 29 for data cleaning and analysis. SPSS Amos version 26 was used to assess aspects of reliability and validity of the latent variables. Finally, descriptive and inferential analysis (hierarchical linear multiple regression) were carried out and the results presented using quantitative metrics such as tables and figures.

## 3.5 Ethical consideration

The study was approved by the University of Education Winneba, Department of Health Administration and Education ethical approval team. The study ensured written consent and anonymity for the study respondents. Prior to the study, respondents were properly informed

about the study through a consent letter sent to their WhatsApp platform before participating in the survey. Respondents who participated in this study therefore did so of their own free will, as there was no incentive attached to responding to the survey questionnaire. Moreover, they were allowed to skip demographic questions that they were not glad to answer.

## 4. Results

### 4.1 Demographic characteristics of respondents

The demographic data of the respondents who participated in this study were explored. The results from the analysis showed that, out of the one thousand and one (1,001) respondents who participated in the survey, 476 representing 47.6% of the study respondents' sex assigned at birth were female and the remaining 52.4% were male. With respect to their age distribution, the minimum and maximum ages of the respondents were 18 and 70 years respectively, with the mean age being 27 years. The majority of the study respondents were found to be Christians (71.1%), followed by Muslims (21.1%), and Traditionalists (4.1%). Most of the study respondents were also found to be first degree holders (74.8%), followed by Masters (15.8%) and Diploma (8.9%). The Akans (52.9%) were found to be the largest ethnic group from which the respondents emanated, followed by the Ga (20.2%) and the Ewe (14.1%). A detailed analysis of the demographic characteristics of the study respondents is presented in Table 1.

### 4.2 Reliability and validity

A preliminary principal component analysis with varimax rotation was carried out to explore the structure and commonalities among the items. The Kaiser-Meyer-Olkin measure of

**Table 1. Demographic characteristics of study respondents.**

| Variables | Values | N | % | MIN | MEAN | MAX | MODE |
|---|---|---|---|---|---|---|---|
| Sex assigned at birth | Female | 476 | 47.6 | | | | |
| | Male | 525 | 52.4 | | | | |
| | Total | 1001 | 100 | | | | |
| Age | | | | 18 | 27.1 | 70 | 27 |
| Religion | Christianity | 743 | 71.1 | | | | |
| | Muslims | 211 | 21.1 | | | | |
| | Traditionalist | 41 | 4.1 | | | | |
| | No Religion | 2 | 0.2 | | | | |
| | No Response | 4 | 0.4 | | | | |
| | Total | 1001 | 100 | | | | |
| Educational level | Diploma | 89 | 8.9 | | | | |
| | First Degree | 749 | 74.8 | | | | |
| | Masters | 158 | 15.8 | | | | |
| | PhD | 5 | 0.5 | | | | |
| | Total | 1001 | 100 | | | | |
| Ethnic background | Akan | 530 | 52.9 | | | | |
| | Ga | 202 | 20.2 | | | | |
| | Ewe | 141 | 14.1 | | | | |
| | Nzema | 85 | 8.5 | | | | |
| | Others | 43 | 4.3 | | | | |
| | Total | 1001 | 100 | | | | |

**Table 2. Fit indices for the model.**

| Test for Exact Fit | | | Additional Fit Measures | | | | |
|---|---|---|---|---|---|---|---|
| $\chi^2$ | df | P | CFI | TLI | GFI | SRMR | RMSEA |
| 684.09 | 129 | 0 | 0.957 | 0.949 | 0.923 | 0.0408 | 0.066 |

sampling adequacy test (MSA) was found to be 0.930 which is above the minimum threshold of 0.80. The Bartlett's Test of sphericity ($\chi^2$ (153) = 13093.690 p = <0.001) was also found to be significant. The extraction of all the items (Table 3) were above the 0.4 minimum extraction as recommended by Beavers et al. [42] with the total variance explained being 72%.

A four-factor confirmatory analysis was then carried out using SPSS Amos version 26 to assess aspects of reliability and validity of the latent variables (i.e. support for the passage of anti-LGBTI legislation, religious beliefs, cultural values, and perceived health implication of LGBTI) measured in the study. The analysis was carried out using the maximum likelihood estimator since this estimator is found to produce equally satisfactory output compared to other estimators for a 7-point Likert scale and also proves to be sufficient and approximates the conditions of the normal distribution, especially with large samples [43]. As indicated in Table 2, crucial aspects of the fit indices values for the model were within the limit of common acceptable ranges [44].

For a construct to be reliable, item loadings must exceed a minimum value of 0.6 as recommended by Bryne [45] and the same is true for the Cronbach's alpha values for the latent variables. To establish convergent validity, Hair et al. [46] inferred that latent variables with a composite reliability (CR) and average variance extracted (AVE) above 0.70 and 0.5 respectively, satisfy or establish convergence validity. From Table 3, it can be observed that except cultural values whose AVE was less than 0.5 (still valid since its CR was greater than 0.70), all of these aspects of reliability and validity elaborated were satisfied.

**Table 3. Assessment of the measurement model.**

| Latent Variable | Items | Commonalities (PCA) | | CFA Loading | Cronbach's alpha (α) | CR | AVE |
|---|---|---|---|---|---|---|---|
| | | Initial | Extraction | | | | |
| Support for the passage of anti-LGBTI laws | SLGBTI1 | 1 | 0.655 | 0.716 | 0.888 | 0.89 | 0.576 |
| | SLGBTI2 | 1 | 0.645 | 0.675 | | | |
| | SLGBTI3 | 1 | 0.667 | 0.747 | | | |
| | SLGBTI4 | 1 | 0.695 | 0.805 | | | |
| | SLGBTI5 | 1 | 0.698 | 0.786 | | | |
| | SLGBTI6 | 1 | 0.704 | 0.816 | | | |
| Religious beliefs | RB1 | 1 | 0.779 | 0.788 | 0.854 | 0.859 | 0.604 |
| | RB2 | 1 | 0.587 | 0.800 | | | |
| | RB3 | 1 | 0.731 | 0.684 | | | |
| | RB4 | 1 | 0.723 | 0.831 | | | |
| Cultural values | CV1 | 1 | 0.631 | 0.618 | 0.789 | 0.789 | 0.484 |
| | CV2 | 1 | 0.693 | 0.710 | | | |
| | CV3 | 1 | 0.549 | 0.673 | | | |
| | CV4 | 1 | 0.589 | 0.773 | | | |
| Perceived health implications | PHI1 | 1 | 0.885 | 0.911 | 0.968 | 0.968 | 0.882 |
| | PHI2 | 1 | 0.899 | 0.923 | | | |
| | PHI3 | 1 | 0.938 | 0.971 | | | |
| | PHI4 | 1 | 0.920 | 0.952 | | | |

**Table 4. HTMT ratio.**

| | SLGBTI | RB | CV | PHI |
|---|---|---|---|---|
| **SLGBTI** | | | | |
| **RB** | 0.548 | | | |
| **CV** | 0.575 | 0.845 | | |
| **PHI** | 0.483 | 0.51 | 0.553 | |

Again, to satisfy discriminant validity using the heterotrait-monotrait (HTMT) ratio of correlations method, it is recommended that the HTMT values be less than 0.9 [47]. As can be seen in Table 4, the discriminant validity of the constructs have also been established.

### 4.3 Support for the passage of anti-LGBTI legislation

The degree of support is presented in Table 5 (values are in percentages except for the mean and standard deviation values). Findings indicate that there was moderately high support for the passage of anti-LGBTI laws (81% of the respondents). This figure (81%) was obtained by adding the average total score of the somewhat agree (SWA), agree (A), and strongly agree (SA) ratings of the six items used to assess the variable. The lower ratings denoting no or less support for the passage of anti-LGBTI legislation; thus, strongly disagree (SD), disagree (D), somewhat disagree (SWD) and neither agree nor disagree (N) cumulative average total score for the six items approximates 19%.

Respondents were again asked to indicate the reasons for which they thought there should be laws against LGBTI and related activities. The results of the study indicated that the majority of the respondents identified health implications (63%) followed by cultural and societal values (62%) and religious reasons (54%) as the reasons for the enactment of anti-LGBTI laws. Twenty-five (25) percent of the respondents also identified western culture as a reason, while seven (7%) indicated none. The respondents were again followed with a question to indicate the single most relevant reason for the passage of anti-LGBTI legislation in the country. As presented in Fig 2 (comparison of reasons and the single most relevant reason), health implications was still identified by a significant number of the respondents (36%), followed by cultural and societal values and religious reasons (24% each). Nine (9) percent of the respondents also recognised western culture as the single most relevant reason, while seven (7) percent indicated no reason.

### 4.4 Perceived health implications of LGBTI and related activities

Whilst it is less easy and likely to question people's religious beliefs and cultural values based on empirical analysis, it is far easy with respect to perceived health implications about some

**Table 5. Support for the passage of anti-LGBTI legislations.**

| Items | SD | D | SWD | N | SWA | A | SA | Mean | SD |
|---|---|---|---|---|---|---|---|---|---|
| SLGBTI1 | 7 | 3.8 | 4.9 | 2.3 | 17.1 | 45.5 | 19.5 | 5.3 | 1.7 |
| SLGBTI2 | 6.9 | 3.2 | 5.5 | 6.9 | 15.4 | 41.4 | 20.8 | 5.3 | 1.7 |
| SLGBTI3 | 4.6 | 4.5 | 6.4 | 4.6 | 20.3 | 41.9 | 17.8 | 5.3 | 1.5 |
| SLGBTI4 | 1.7 | 4.2 | 6.6 | 5.3 | 17.1 | 40.5 | 24.7 | 5.5 | 1.7 |
| SLGBTI5 | 1.9 | 3.3 | 5.2 | 4.1 | 18.5 | 41.5 | 25.6 | 5.6 | 1.4 |
| SLGBTI6 | 2.9 | 4.5 | 5.8 | 6.6 | 21.2 | 42.7 | 16.4 | 5.32 | 1.5 |
| Average Total Score | 4.20% | 3.90% | 5.70% | 5% | **18.20%** | **42.20%** | **21%** | 5.4 | 1.6 |

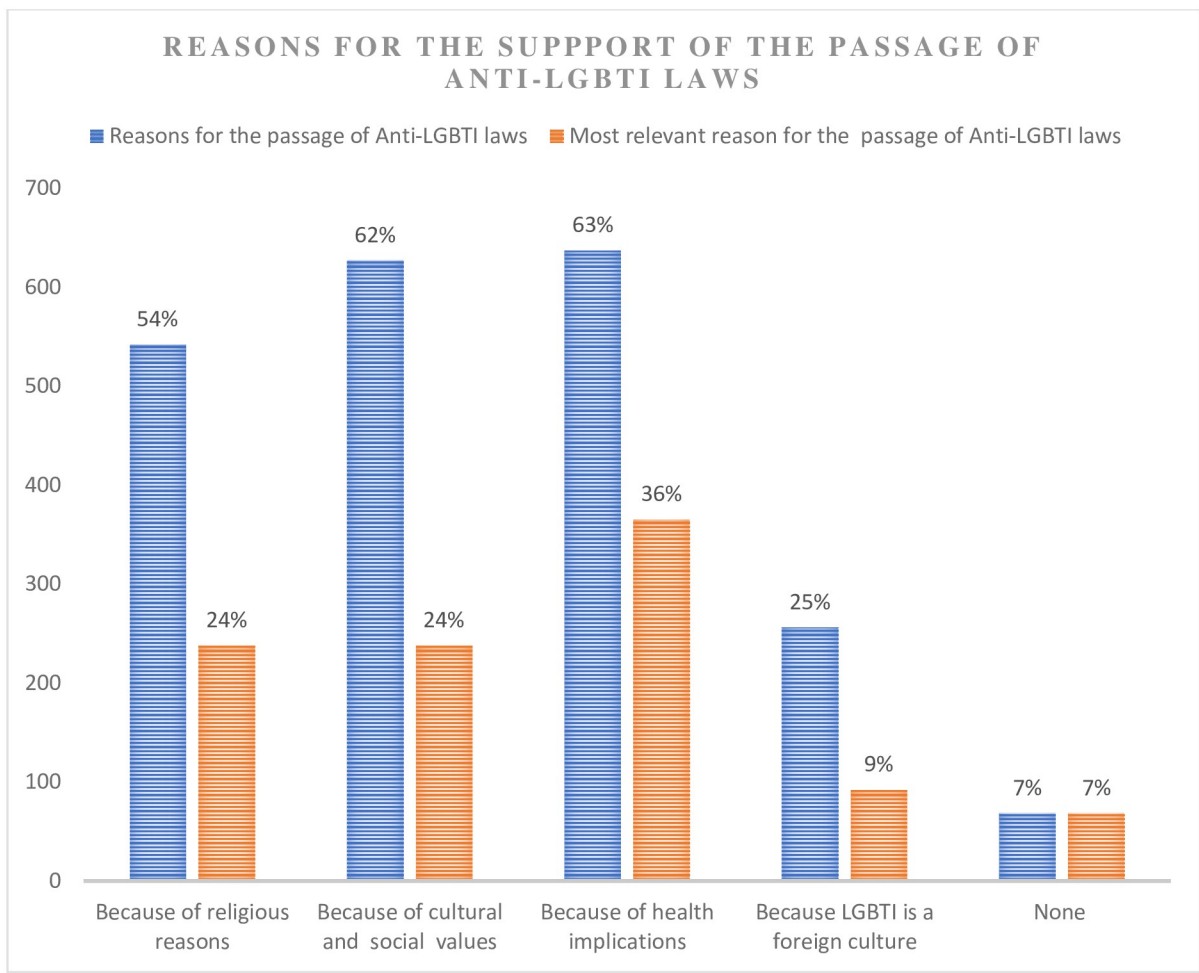

**Fig 2. A figure showing the comparison between reasons and most relevant reason for the support of the passage of anti-LGBTI legislation.**

phenomenon or activity. The perceived health implications of LGBTI, which was assessed with four Likert scale (7-point) items were therefore analysed. Strongly disagree, disagree and somewhat disagree were treated as "disagree." Similarly, strongly agree, agree and somewhat agree were also treated as "agree." As presented in Fig 3, about 49% of the study respondents were found to hold some perceptions about the health implications of LGBTI that are not scientifically legitimate. This is alarming given the context that respondents in the study have some significant level of education.

## 4.5 Predictors of support of anti-LGBTI legislation

One of the objectives of the study was to identify non-physical factors that predict support for the passage of anti-LGBTI laws among tertiary-level students in Ghana. Hence, a hierarchical multiple regression was carried out to test if cultural values, religious beliefs, and perceived health implications of LGBTI predict students support for anti-LGBTI laws when age and sex assigned at birth are controlled. Since the sample size was significantly larger than 200 (i.e., 1001), the associated normality assumptions of regression analysis were not crucial, as the central limit theorem is invoked to ensure that the critical aspect of "residual distribution"

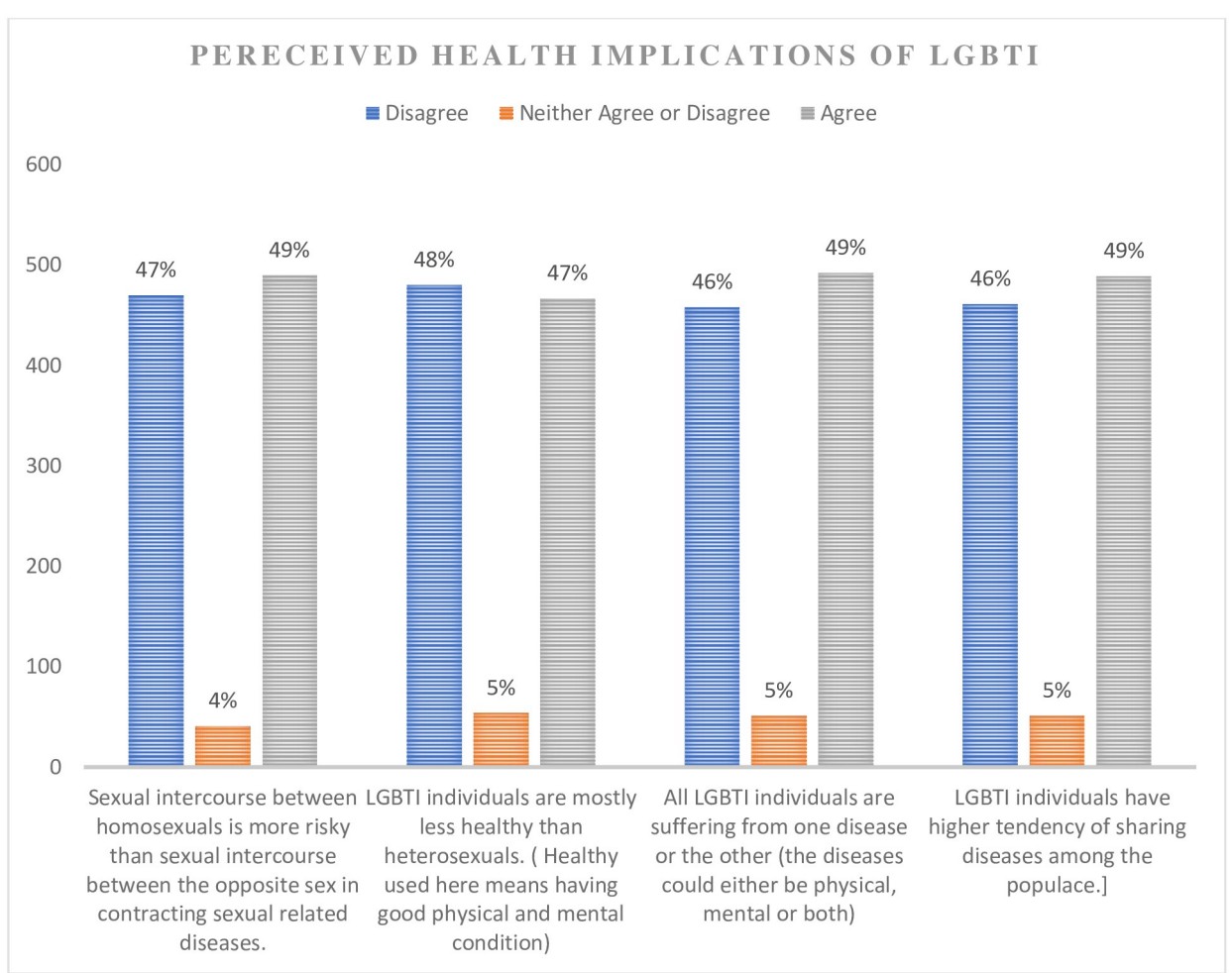

**Fig 3. A figure showing the respondents' responses on the perceived health implications of LGBTI and related activities.**

approximate normality [48]. The Durbin-Watson test run alongside the hierarchical multiple regression indicated no serious autocorrelation problem's (d = 1.76) existence in the regression residuals.

The results of the regression analysis (Table 6) depicted that predictors accounted for 29.3% of the variance in support for the passage of anti-LGBTI legislation ($R^2 = 0.293$, F (3, 995) = 139.18), p = < .001) when sex and age are controlled. Exploring the unique contribution of the individual predictors, the results of the study showed that three of the predictors (Table 7);

**Table 6. Model fitness.**

| Model | R | $R^2$ | Adjusted $R^2$ | Change statistics | | | | |
|---|---|---|---|---|---|---|---|---|
| | | | | $R^2$ change | F change | df1 | df2 | p |
| 1 | .086a | 0.007 | 0.005 | 0.007 | 3.744 | 2 | 998 | 0.024 |
| 2 | .548b | 0.301 | 0.297 | 0.293 | 139.18 | 3 | 995 | < .001 |

[a]Predictors: age and sex

[b]Predictors: age, sex, perceived health implication of LGBTI, religious belief and cultural values.

**Table 7. Model coefficients.**

| Model | Predictors | B | SE | Beta | t | p | Tolerance | VIF |
|---|---|---|---|---|---|---|---|---|
| 1 | Constant | 6.384 | 0.38 | | 16.788 | < .001 | | |
| | Age | -0.037 | 0.014 | -0.085 | -2.691 | 0.007 | 0.995 | 1.005 |
| | Sex(female) | 0.023 | 0.078 | 0.009 | 0.299 | 0.765 | 0.995 | 1.005 |
| 2 | Constant | 2.264 | 0.392 | | 5.77 | < .001 | | |
| | Age | -0.009 | 0.012 | -0.021 | -0.803 | 0.422 | 0.979 | 1.021 |
| | Sex(female) | 0.031 | 0.066 | 0.012 | 0.468 | 0.640 | 0.994 | 1.006 |
| | PHI | 0.157 | 0.02 | 0.247 | 8.002 | < .001 | 0.738 | 1.355 |
| | RB | 0.225 | 0.046 | 0.189 | 4.909 | < .001 | 0.472 | 2.117 |
| | CV | 0.272 | 0.048 | 0.218 | 5.659 | < .001 | 0.474 | 2.111 |

perceived health implication of LGBTI ($\beta$ = 0.247, t = 8.002, p = < .001) followed by cultural values ($\beta$ = 0.218, t = 5.659, p = < .001) and religious beliefs ($\beta$ = 0.189, t = 4.909, p = < .001) independently and positively predict support for passage anti-LGBTI legislation even when age and sex is controlled. One can therefore infer from the results that individual student religious beliefs, cultural values, and perceived health implications of LGBTI and related activities are likely to positively influence people's support for the passage of anti-LGBTI legislation in the country (Ghana), just as identified in the results of the descriptive statistics of the study.

## 5. Discussion

The use of both descriptive and inferential statistics was preferred to provide a greater depth of understanding to the study's wide audience. According to the study results, about 81% of the study respondents, to some extent, are always ready and in support of the passage of any related anti-LGBTI legislation in the country. This implies that only about 15% of the population may not support the passage of any anti-LGBTI-related legislation in the country at a given point in time. Referencing the fact that the study respondents were tertiary-level students and are supposed or anticipated to be more tolerable, more exposed to modernity and human right notions such that such result would not have been obtained, it does signify that in practice, greater number of the populace (above 85%) are likely to support the passage of any anti-LGBTI related legislation since we do not expect local indigenous or people with less education and are more inclined to cultural values and religious indoctrinations to deviate from the trend of the results. This result thus, confirms to an extent the continuous arguments made in the various media platforms in the country that Ghanaians are against LGBTI and related activities.

The study result is in consistence with the findings of other studies [6, 7] and argument of some African scholars and leaders such as President Museveni on the reasons why LGBTI and related activities could not be welcomed in African societies [8, 17]. Reasons accounting for the support of the passage of anti-LGBTI and related legislation were identified as the health implications of LGBTI and related activities (63%), societal and cultural values (62%), religious reasons (54%), and foreign culture notion (25%). As to whether these reasons are substantial enough to pass legislation in a democratic and pluralistic religious state, this is an issue of debate. Whilst less could be argued against societal and cultural values reason given as Africans with a unique identity and cultural heritage, we doubt it may be true for the other reasons cited. The study, for instance, revealed religious reasons, health implications, and foreign culture notion to account for 57% (out of 93%) of the most important or crucial reasons for which respondents support the passage of anti-LGBTI and related legislation (see Fig 1). These

three reasons, as mentioned earlier, may or may not be substantial enough for a law to be passed in a democratic state. In an attempt to make inference on these reasons identified, perhaps some questions for policymakers to address include: is it rational and ideal for a law to be passed in a pluralistic religious and democratic state because an activity is against some religious doctrine, is a foreign culture, and is perceived to have negative health implications but has little or no empirical evidence?

It can be seen from Fig 2 that most of the perceived health implications of LGBTI and related activities are not scientifically legitimate. To the best of our knowledge, there is no substantive empirical evidence that suggests that heterosexuals are healthier than the homosexuals nor are there evidence to support a claim that LGBTI individuals are more likely to share diseases easily among the populace than the heterosexuals. What is probably of concern and demeaning is why respondents who have some appreciable level of education may have these notions. Again, a far worse scenario may be expected to be observed among the population with less education with regard to these perceptions. These misconceptions, therefore, need to be addressed.

The inferential analysis indication of perceived health implications, cultural values and religious belief as predictors of support for the passage of anti-LGBTI legislation is just a confirmation of the descriptive result of the study and other related studies' claims of reasons for legislation and discrimination against LGBTI individuals [6, 7]. All the predictors may also account for the way and manner in which LGBTI individuals are sometimes maltreated or discriminated against in the country and beyond [13, 21, 23].

In the absence of these three predictors, age (B = -0.085, p = 0.007) seems to negatively influence the support for the passage of anti-LGBTI Legislation implying older students are less likely to fully support anti-LGBTI legislation. This appears to be contradictory to the general notion that young people are more likely to accept LGBTI individuals and hence are less likely to support an initiative against them. Given the study setting, a possible explanation to this perceived contradiction may be the fact that the older students in the study may have advance degrees and may be more abreast with human right and equality issues and hence less support for anti-LGBTI legislation. Again, from the results of the study, sex assigned at birth appears to have no influence on people's support for anti-LGBTI legislation. The reason could be that the issue at hand may have more elements of social constructs than biology. An issue characterised by some authorities and leaders as a taboo in some societies [29] is less likely to be influenced by sex.

The study's findings could be better understood in light of the three theories or concepts that underpinned or informed the conceptual framework of the study. First of all, as posited by the African Heritage Theory, Africans have a specific culture with a unique set of values that differs from the rest of the world. LGBTI and related activities, such as same-sex marriage, are argued to be against the norms and values of most communities in Africa. Though such claims have been challenged [14–16], it appears to be a popular argument. The basis of most laws against LGBTI and related legislation in the African region has always had the basic premise of promoting and protecting "cultural values" as in the case of the current passed Ugandan anti-LGBT law and the proposed anti-LGBTI law before Ghana's Parliament, dubbed the "promotion of proper human sexual rights and Ghana's family values bill". The banning of some homosexuals from some African communities [21], the acclamation of LGBTI as a "taboo" by some traditional leaders such as Prof. Nii Odaifio III [29], and the calling by some African presidents [8] on the Western World to mind their businesses and respect the culture of African societies with regards to legislation concerning LGBTI are all reflections on the fact that LGBTI and related activities are predominantly seen as alien to the culture of African societies and therefore could be seen among activities that fall outside the boundaries of African culture

when viewed through the lenses of the African heritage theory. Cultural values as a reason and predictor of support for the passage of anti-LGBTI legislation in the study findings is therefore not surprising.

Secondly, the health belief model emphasises that people are likely to take urgent actions when they perceive that an action or set of actions has the tendency to have precarious health implications for their lives and those of their loved ones. The unscientific perceptions of negative health implications of LGBTI and related activities in the findings of the study through the lenses of the health belief model might partly explain why the majority of people are in support of the passage of anti-LGBTI and related legislations.

The last but not the least is the religious "concept of sin" as defined in the various religious books forming the core of religious beliefs, which in part have been used as the basis for determining morality by the various influential religions such as Christianity and Islam. Most Christians and Muslims are of the view that LGBTI and related activities such as same-sex marriage and sexual intercourse are great sins (immoral) and hence not to be entertained in society. To the leaders of these major religions, such activities would incur God's wrath on humanity and consequently lead to the destruction of the world, as in the Christian perspective led to the destruction of "Sodom and Gomora" in the Bible [21, 49]. This might have explained why some of the Christians are threatening politicians in the country (Ghana) to pass a law against LGBTI or else they will campaign against the politicians [49]. This might also explains why most Islamic States across the world, in particular, are among the States with the harshest laws against LGBTI and related activities [4]. The identification of religious belief as the reason and one of the predictors for the support of the passage of anti-LGBTI and related legislation in the study is thus explained to an extent, as Christianity and Islamic religion account for about 92% of the study respondents.

Given the political power held by traditional authorities and the leaders of religious groups such as Christianity and Islam due to their large numbers, it would not be surprising for legislation to be passed in the country in the near future due to their influence alone than to talk about their protest and agitations. Even so, the factors identified in the study for the support of anti-LGBTI legislation based on the perspective of tertiary-level students in the country would have to be continually weighed against the equally popular opposing "human rights" views by various international organisations and human rights activists to guide and ensure the optimum outcome of LGBTI and related policies.

## 5.1 Limitations and future research

Even though conscious efforts were put in place to ensure a robust study, one cannot infer that the study is without any methodological limitations. For instance, the study only considered the students in the country who were active on the WhatsApp platform at the time of the data collection. The study also does not include those who have no tertiary education, and since a significant number of the population is without tertiary education, the generalisation of the study must be done to a certain limit. Along with this, the study also used the convenience sampling method, which makes the study susceptible to the disadvantages of non-probability sampling methods even though the sample size is significantly adequate for generalisation.

In the light of the above, the study recommends a study with a specific focus on people without tertiary education in the country to complement the findings of this study. Future studies must deploy traditional and fully probabilistic sampling methods as well as test the knowledge of students on human rights and freedoms and sexuality laws in the 1992 Constitution of Ghana to better widen the scope of understanding of the issue under study in the study setting.

### 5.2 Policy implications

This is an essential study for policymakers in this crucial moment where there are regular LGBTI and related legal issues being discussed in the African region and beyond. It is also relevant at the moment, where there is an anti-LGBTI-related bill before the Parliament of Ghana (the study setting) for consideration. The study does not only provide evidence of support for the passage of anti-LGBTI and related laws among the students but also gives details to the reasons behind the support. Policymakers and other influential stakeholders' perspectives on the issues of LGBTI and related legal issues should be shaped by the detailed results of the study rather than an aspect of the study results. More specifically, it is incumbent on policymakers based on the findings of the study to do an in-depth analysis as to whether the reasons identified for the support of anti-LGBTI and related issues are indeed imperative in making legislation, especially in a democratic state. The study also highlights the need for policymakers and various stakeholders to create awareness and educate the public on the various perceptions about LGBTI and related activities that are not scientifically legitimate.

Finally, the study results to some extent imply the need to sensitise and educate the Ghanaian populace about the provisions made in the 1992 Constitution of Ghana with respect to human rights and freedoms and sexuality, as well as the provisions in the 1948 Universal Declaration of Human Rights of the United Nations, when it comes to the passage of legislation regarding or related to human sexuality and right.

## Supporting information

**S1 File. Implied correlations of the latent variables.**
(XLSX)

**S2 File. Data.**
(XLSX)

**S3 File. Regression output.**
(DOCX)

**S1 Appendix. A list of items used in measuring the study variables.**
(DOCX)

## Acknowledgments

We acknowledge and appreciate the unrelenting technical support given by Prof. Lucia Savadori and Rawlings Ntassah both at the University of Trento, Italy throughout the entire research process.

## Author Contributions

**Conceptualization:** Francis Acquah.

**Data curation:** Francis Acquah, Charles Owusu-Aduomi Botchwey, Prince Owusu Adoma, Emmanuel Kumah.

**Formal analysis:** Francis Acquah, Charles Owusu-Aduomi Botchwey, Prince Owusu Adoma.

**Investigation:** Francis Acquah, Prince Owusu Adoma.

**Methodology:** Francis Acquah, Prince Owusu Adoma.

**Project administration:** Charles Owusu-Aduomi Botchwey, Emmanuel Kumah.

**Resources:** Francis Acquah, Charles Owusu-Aduomi Botchwey, Prince Owusu Adoma.

**Software:** Francis Acquah, Charles Owusu-Aduomi Botchwey.

**Supervision:** Charles Owusu-Aduomi Botchwey.

**Validation:** Charles Owusu-Aduomi Botchwey.

**Visualization:** Francis Acquah, Emmanuel Kumah.

**Writing – original draft:** Francis Acquah.

**Writing – review & editing:** Francis Acquah, Charles Owusu-Aduomi Botchwey, Prince Owusu Adoma, Emmanuel Kumah.

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
