## [Decision Letter · Decision Letter 0]

10 Apr 2023

PONE-D-23-03818Lesbian, Gay, Bisexual, Transgender, Intersex, and Related Legal Issues Arising in Africa: Insights from the Graduates in GhanaPLOS ONE

Dear Dr. Acquah,

Thank you for submitting your manuscript to PLOS ONE. After careful consideration, we feel that it has merit but does not fully meet PLOS ONE’s publication criteria as it currently stands. Therefore, we invite you to submit a revised version of the manuscript that addresses the points raised during the review process.

We look forward to receiving your revised manuscript.

Kind regards,

Godfred Matthew Yaw Owusu

Academic Editor

PLOS ONE

3. Please include a separate caption for each figure in your manuscript

Reviewers' comments:

Reviewer's Responses to Questions

**Comments to the Author**

1. Is the manuscript technically sound, and do the data support the conclusions?

Reviewer #1: Partly

Reviewer #2: Partly

2. Has the statistical analysis been performed appropriately and rigorously? 

Reviewer #1: Yes

Reviewer #2: Yes

3. Have the authors made all data underlying the findings in their manuscript fully available?

Reviewer #1: Yes

Reviewer #2: Yes

4. Is the manuscript presented in an intelligible fashion and written in standard English?

Reviewer #1: No

Reviewer #2: Yes

5. Review Comments to the Author

Reviewer #1: 1. The topic is relevant at this time and thus worth writing on. However it appears to be misleading. Perhaps it could be reframed to reflect the write up as the study seems to establish the perception of tertiary students on LGBTI and its Legalization. The title '... legal issues arising' appears to indicate a legal digestion of the bill that has been presented before the parliament of Ghana.

2. The problem is unclear. What is the gap in literature? What is the issue about a country passing a law on LGBTI? It appears the study sort to ascertain the perception of tertiary students on LGBTI and the attempt to legalize it in Ghana. The authors should concentrate on looking at the perception tertiary students have on LGBTI and the on the attempt to criminalize it. The query of the authors as to whether religion, health and cultural beliefs are the adequate in criminalizing LGBT is a non-starter as these form the basis of laws the world over.

3.The authors appear to be confused about their respondents. The word Graduate student refers to people pursuing a second or third degree. In the paper, they refer to their respondents as graduate students. Which is different from the word graduate in the title. The also indicated some of their respondents were yet to graduate. This makes them students. Thus the use of the word Graduate in the title does not correspond with the actual respondents who took part in the study. Perhaps the appropriate word to use is Tertiary level students.

4. The authors mentioned two theories that informed the study but failed to demonstrate how these theories provide any logical explanation to their findings. The theories were not used to explain the conceptual framework developed. Neither did the theories inform the questionnaire that was developed and used to collect data.

Reviewer #2: The manuscript borders on a topical issue under discussion globally in various religious and cultural settings. And although the topic seems to be topical in the media, the authors do not clearly define the issue gap based on the literature, to clearly show what the intended contribution to the literature is. The study mentions that there are a number of studies in this area, yet no study has been conducted on people’s opinion. Questions such as what makes an opinion study relevant to expand knowledge in this area and how does this contribute to the existing state of literature in this area is not clearly articulated.

The choice of graduate students as the subject of study is not properly justified, to understand why their perspective matters and how their perspective would expand knowledge under this gap. Also, these students were explained to be from the Central and southern parts of Ghana. The reason for the choice was not explained and justified.

The methodology used under the study is appropriate, yet a pretested and existing scales for assessing the factors under review would have greatly enriched the reliability of the constructs.

The discussion of the results has not been rigorously done, to draw out the theoretical implications based on the theory employed in the study. Also, the discussion section gives a general summary of findings, without an in-depth reflection

with the existing empirics. Therefore, the paper lacks insights into the empirical implications of the findings beyond just the statistics discussed.

6. PLOS authors have the option to publish the peer review history of their article (what does this mean?). If published, this will include your full peer review and any attached files.

Reviewer #1: No

Reviewer #2: No

---

## [Author Response · Author response to Decision Letter 0]

25 Apr 2023

Dear Editor,

RESPONSE TO REVIEWERS’ COMMENTS

We acknowledge the complexity and various schools of taught on the topic in real life and hence appreciate the reviewers’ comments and suggestions in improving the content of the paper.

We have addressed all the issues raised and believe the current version of the paper meets the journal’s publication requirements.

We are very grateful for this wonderful opportunity to publish in your highly esteemed journal.

Please find below our responses to the issues raised:

Reviewer #1 

Comment 1: The topic is relevant at this time and thus worth writing on. However, it appears to be misleading. Perhaps it could be reframed to reflect the write up as the study seems to establish the perception of tertiary students on LGBTI and its Legalization. The title '... legal issues arising' appears to indicate a legal digestion of the bill that has been presented before the parliament of Ghana.

Response: Please the topic has been reframed and it currently reads as follows: Lesbian, Gay, Bisexual, Transgender, Intersex, and its Legalisation in Africa: Insights from Tertiary-Level Students in Ghana

Comment 2: The problem is unclear. What is the gap in literature? What is the issue about a country passing a law on LGBTI? It appears the study sort to ascertain the perception of tertiary students on LGBTI and the attempt to legalize it in Ghana. The authors should concentrate on looking at the perception tertiary students have on LGBTI and the on the attempt to criminalize it. The query of the authors as to whether religion, health and cultural beliefs are the adequate in criminalizing LGBT is a non-starter as these form the basis of laws the world over.

Response: Please the problem statement has further been clarified in the “focus of the present study” section. Please see page 8, lines 206-214 of the revised manuscript for this information. 

Comment 3: The authors appear to be confused about their respondents. The word Graduate student refers to people pursuing a second or third degree. In the paper, they refer to their respondents as graduate students. Which is different from the word graduate in the title. The also indicated some of their respondents were yet to graduate. This makes them students. Thus, the use of the word Graduate in the title does not correspond with the actual respondents who took part in the study. Perhaps the appropriate word to use is Tertiary level students

Response: Please at the time of data collection, the participants had completed university though not officially done their graduation at their respective universities hence the reference of respondents as graduates nevertheless the term has been appropriately changed to the inclusive reviewer’s recommendation as “tertiary-level students”. 

Comment 4: The authors mentioned two theories that informed the study but failed to demonstrate how these theories provide any logical explanation to their findings. The theories were not used to explain the conceptual framework developed. Neither did the theories inform the questionnaire that was developed and used to collect data.

Response: Please the literature together with the theories informed the conceptual framework. The three predictors’ cultural values, perceived health implication and religious belief were informed by the African Heritage Theory, Health Belief Model and Religious concept of sin respectively. Due to conciseness in the writing of the discussion section we fail however to delineate how the theories provide any logical explanation to the findings of the study as mentioned by the reviewer. These comments have been addressed in the revised manuscript in the discussion section (please see lines 522-567) together with some amendment in “Theoretical and Conceptual Background of the Study” section

Reviewer #2 

Comment 1: The manuscript borders on a topical issue under discussion globally in various religious and cultural settings. And although the topic seems to be topical in the media, the authors do not clearly define the issue gap based on the literature, to clearly show what the intended contribution to the literature is. The study mentions that there are a number of studies in this area, yet no study has been conducted on people’s opinion. Questions such as what makes an opinion study relevant to expand knowledge in this area and how does this contribute to the existing state of literature in this area is not clearly articulated.

Response: Please the problem statement has been further clarified in the “focus of the present study” section (Please see page 8, lines 206-214). Reviewer; The study mentions that there are a number of studies in this area, yet no study has been conducted on people’s opinion.” We indicated that “some aspects of the subject of interest in the country have been studied. As cited, these include studies such as “Homophobic Violence in Ghana”, “Perceptions of the Youth towards Homosexuality in Ghana,” and a few others. We did not necessarily mean “opinion studies specifically on LGBTI legislation.” Meanwhile, a greater distinction has been made in the updated manuscript. 

Comment 2: The choice of graduate students as the subject of study is not properly justified, to understand why their perspective matters and how their perspective would expand knowledge under this gap. Also, these students were explained to be from the Central and southern parts of Ghana. The reason for the choice was not explained and justified.

Response: Please the choice of graduate students as the subject of study is further clarified and justified in the updated manuscript in the “focus of the present study” section. Tertiary-level students are appropriate for the study because we consider them to be more tolerable and abreast with the current LGBTI issues in Ghana. Detailed justification for our choice of students could be found from line 222 to 228 of the revised manuscript. 

The rationale for recruiting students from the central and southern parts of Ghana is that these geographical areas capture almost all of the ethnic groups in the country. Thus, the choice ensured adequate representation of the various ethnic groups in the country. Please see lines 270 and 271 for the addition of this information to the revised manuscript. 

Comment 3: The methodology used under the study is appropriate, yet a pretested and existing scales for assessing the factors under review would have greatly enriched the reliability of the constructs.

Response: We wish we could have gotten pretested and existing scales that also take into account the nature of the study setting to assess the factors in the study. Unfortunately, such scales were not available.

Comment 4: The discussion of the results has not been rigorously done, to draw out the theoretical implications based on the theory employed in the study. Also, the discussion section gives a general summary of findings, without an in-depth reflection

with the existing empirics. Therefore, the paper lacks insights into the empirical implications of the findings beyond just the statistics discussed. 

Response: Please the updated manuscript now delineates how the theories provide logical explanation to the findings of the study and possible future implications. Please see lines 523-568 of the revised manuscript for this revision. 

Thank you

---

## [Decision Letter · Decision Letter 1]

12 Jun 2023

Lesbian, Gay, Bisexual, Transgender, Intersex, and its Legalisation in Africa: Insights from Tertiary-Level Students in Ghana

PONE-D-23-03818R1

Dear Dr. Acquah,

We’re pleased to inform you that your manuscript has been judged scientifically suitable for publication and will be formally accepted for publication once it meets all outstanding technical requirements.

Kind regards,

Godfred Matthew Yaw Owusu

Academic Editor

PLOS ONE

Additional Editor Comments (optional):

Reviewers' comments:

Reviewer's Responses to Questions

**Comments to the Author**

1. If the authors have adequately addressed your comments raised in a previous round of review and you feel that this manuscript is now acceptable for publication, you may indicate that here to bypass the “Comments to the Author” section, enter your conflict of interest statement in the “Confidential to Editor” section, and submit your "Accept" recommendation.

Reviewer #1: All comments have been addressed

2. Is the manuscript technically sound, and do the data support the conclusions?

Reviewer #1: Yes

3. Has the statistical analysis been performed appropriately and rigorously? 

Reviewer #1: Yes

4. Have the authors made all data underlying the findings in their manuscript fully available?

Reviewer #1: Yes

5. Is the manuscript presented in an intelligible fashion and written in standard English?

Reviewer #1: Yes

6. Review Comments to the Author

Reviewer #1: (No Response)

7. PLOS authors have the option to publish the peer review history of their article (what does this mean?). If published, this will include your full peer review and any attached files.

Reviewer #1: **Yes: **Anita Asiwome Baku

---

## [Editor Report · Acceptance letter]

26 Jun 2023

PONE-D-23-03818R1 

*Lesbian, Gay, Bisexual, Transgender, Intersex, and its Legalisation in Africa: Insights from Tertiary-Level Students in Ghana*

Dear Dr. Acquah:

I'm pleased to inform you that your manuscript has been deemed suitable for publication in PLOS ONE. Congratulations! Your manuscript is now with our production department. 

Kind regards, 

on behalf of

Dr. Godfred Matthew Yaw Owusu 

Academic Editor

PLOS ONE